# Effect of Ultra-High Pressure Sintering and Spark Plasma Sintering and Subsequent Heat Treatment on the Properties of Si_3_N_4_ Ceramics

**DOI:** 10.3390/ma15207309

**Published:** 2022-10-19

**Authors:** Xiaoan Lv, Xianhui Li, Junwei Huang, Changchun Ge, Qi Yu

**Affiliations:** 1School of Materials Science and Engineering, University of Science and Technology Beijing (USTB), Beijing 100083, China; 2Hebei Normal University for Nationalities, Chengde 067000, China; 3Luoyang Institute of Science and Technology, Luoyang 471023, China; 4Luoyang Bearing Research Institute Co., Ltd., Luoyang 471023, China

**Keywords:** coarse β-Si_3_N_4_ powders, spark plasma sintering, ultra-high pressure sintering, densification, mechanical property, thermal conductivity

## Abstract

In this study, coarse Beta silicon nitride (β-Si_3_N_4_) powder was used as the raw material to fabricate dense Si_3_N_4_ ceramics using two different methods of ultra-high pressure sintering and spark plasma sintering at 1550 °C, followed by heat treatment at 1750 °C. The densification, microstructure, mechanical properties, and thermal conductivity of samples were investigated comparatively. The results indicate that spark plasma sintering can fabricate dense Si_3_N_4_ ceramics with a relative density of 99.2% in a shorter time and promote α-to-β phase transition. Coarse β-Si_3_N_4_ grains were partially fragmented during ultra-high pressure sintering under high pressure of 5 GPa, thereby reducing the number of the nucleus, which is conducive to the growth of elongated grains. The UHP sample with no fine α-Si_3_N_4_ powder addition achieved the highest fracture strength (822 MPa) and fracture toughness (6.6 MPa·m^1/2^). The addition of partial fine α-Si_3_N_4_ powder facilitated the densification of the SPS samples and promoted the growth of elongated grains. The β-Si_3_N_4_ ceramics SPS sintered with fine α-Si_3_N_4_ powder addition obtained the best comprehensive performance, including the highest density of 99.8%, hardness of 1890 HV, fracture strength of 817 MPa, fracture toughness of 6.2 MPa·m^1/2^, and thermal conductivity of 71 W·m^−1^·K^−1^.

## 1. Introduction

Silicon nitride (Si_3_N_4_) ceramics are important structural materials applied for engine components, ball bearings, cutting tools, and heat exchangers for a long time owing to their excellent mechanical properties, high chemical stability, superior wear, and thermal resistance [1,2,3]. Especially, Si_3_N_4_ ceramics shows intrinsic thermal conductivity of about 200 W·m^−1^·K^−1^ at room temperature [4]. Therefore, Si_3_N_4_ ceramics with high strength and thermal conductivity are considered promising candidates for substrate materials and heat sinks in hybrid motor vehicles, industrial robots, advanced trains, and aircraft [5,6,7].

Si_3_N_4_ ceramics are generally sintered by a liquid phase sintering mechanism added to sintering aids to form polycrystals, owing to their strong covalent bond and low diffusivity [8]. Numerous kinds of research have shown that increasing densification was crucial to improving both the thermal and mechanical properties of Si_3_N_4_ ceramics [9,10]. Moreover, large grains can improve the heat transmission efficiency, and the low content of inter-granular secondary phase and lattice defects can reduce phonon scattering to improve thermal conductivity [11,12,13].

High thermal conductivity Si_3_N_4_ ceramic can be produced by hot-pressing, gas press sintering, and sintering reaction-bonded Si_3_N_4_ (SRBSN) [12,14,15]. Wang et al. [16] prepared ceramics with a thermal conductivity of 117 W·m^−1^·K^−1^ by gas pressure sintering using a binary non-oxide sintering additive ZrSi_2_-MgSiN_2_. Zhou et al. [5] used a modified SRBSN method post-sintered at 1900 °C for 60 h to prepare a Si_3_N_4_ ceramic with a record-high thermal conductivity of 177 W·m^−1^·K^−1^ and a high fracture toughness of 11.2 MPa·m^1/2^. However, high thermal conductivity Si_3_N_4_ ceramic fabricated by the above methods involved a high sintering temperature and long annealing time. From the viewpoint of cost, it is desirable to develop a high thermal conductivity Si_3_N_4_ ceramic with relatively low temperature and within a short period of time.

Spark plasma sintering (SPS) is a novel technique that facilitates the sintering of metals, plastics, ceramics, and composites at low temperatures with short sintering times [17,18,19]. Instead of an external heating source of hot pressing, SPS uses a pulsed, direct current passing through the electrically conducting die and the sample itself, while the spark plasma generates internal localized heating to promote material transfer and accelerates localized reactions as well as densification [20]. Peng et al. [21] achieved Si_3_N_4_ ceramics with a thermal conductivity of 100 W·m^−1^·K^−1^, employing SPS at 1600 °C for 12 min. Chunping Yang et al. [22] first sintered the silicon nitride ceramics with added crystalline species using SPS to densify it, and then subjected the samples to high-temperature heat treatment to obtain high-thermal-conductivity silicon nitride ceramics.

Ultra-high pressure (UHP) sintering is another fast-sintering technique similar to SPS that can heat and cool rapidly [23,24,25]. The powder particles can be in close contact under ultra-high pressure, which is beneficial to the densification of the sintered sample, lowering the sintering temperature and shortening the sintering time. Ultra-high pressure sintering Si_3_N_4_ ceramics with a thermal conductivity of 64.6 W·m^−1^·K^−1^ were prepared under a pressure of 5 GPa at 1490 °C for 60 min, which is attributed to the higher density and small grain size [9]. However, due to the small size of UHP samples, the fracture strength and fracture toughness are rarely tested.

Since fine powder has a more significant sintering driving force, Si_3_N_4_ ceramics are generally produced from fine α-Si_3_N_4_ powders or fine β-Si_3_N_4_ powders as raw material, but the coarse β-Si_3_N_4_ powders are rarely directly used as raw powders or added as a seed crystal, although it has higher thermal conductivity than α-Si_3_N_4_ powders. In the present study, the coarse β-Si_3_N_4_ powders were used as raw material to prepare dense Si_3_N_4_ ceramics firstly by ultra-high pressure sintering and spark plasma sintering. Then the sintered samples were heat treated to eliminate the defects and promote grain growth. The effects of the sintering process, additive amount, and heat treatment on densification, phase transition, microstructure, mechanical properties, and thermal properties of Si_3_N_4_ ceramic were compared. Moreover, the researchers reported that the addition of fine α-Si_3_N_4_ diluents was found to affect the properties by enhancing the density.

## 2. Materials and Methods

High purity β-Si_3_N_4_ powder (particle size: 1.8–2.2 μm, oxygen content: 0.8%; Anyang JSH New Material Co., Ltd., Anyang, China) and α-Si_3_N_4_ powder (particle size: 0.8 μm, oxygen content: 2.2%; Anyang JSH New Material Co., Ltd., Anyang, China) were used as raw powders (details are provided in Table 1). Y_2_O_3_ and MgO (purity > 99.99%, Qinhuangdao Eno High-Tech Material Development Co., Ltd., Qinhuangdao, China) with an average size of 1 μm were the sintering additives. The sample composition is listed in Table 2. The morphology of the raw material powders is illustrated in Figure 1.

Through the use of ZrO_2_ balls and alcohol as the milling medium for 4 h, all powders were thoroughly blended. The slurries were dried in an air oven for 12 h. The combined powders were sieved to a 100-mesh size to prepare for sintering.

For SPS experiments, graphite dies of 20 mm diameter containing 3 g of powders were put inside the SPS 1050 apparatus (Sumitomo Coal Mining Co., Ltd., Tokyo, Japan) after being installed. The samples were sintered in a vacuum environment at 1550 °C for 3 min with a 40 MPa uniaxial stress.

For UHP sintering experiments, all samples were synthesized by using an HT-III cubic press system (Zhengzhou Tianhong Automation Technology Co., Ltd., Zhengzhou, China). A total of 3 g of raw mixed ingredients were cold pressed at 5 MPa into a mold with a diameter of 18 mm, then put into a molybdenum cup. In the meantime, all of the parts were placed inside a 56 × 56 × 56 mm^3^ pyrophyllite composite cavity for the UHP sintering. Figure 2b depicts the cross-section schematic of the sample assembly, and the physical disassembled image is shown in Figure 2c. Then, the pyrophyllite composite cavity was loaded into a six-anvil press for UHP experiments, as shown in Figure 2a. The sintering process curve is shown in Figure 2d. First, 5 GPa pressure is applied to the sample, and then the sample is sintered by heating with an electric current, and after holding the pressure at 1550 °C for 15 min, the pressure is unloaded after cutting off the power to cool down to room temperature.

Subsequently, all the spark plasma sintered samples and ultra-high pressure sintered samples were heat-treated at 1750 °C for 3 h under 4 MPa nitrogen pressure.

After being heat-treated, all samples were cut, polished, and purified for measurements. The densities of the samples were determined using Archimedes’ method. Through X-ray diffraction, the phase compositions of the ceramics were identified (XRD, SmartLab X-ray diffractometer, Japanese science company, Tokyo, Japan). Scanning electron microscopy (SEM, ZEISS, LEO1450, Oberkochen, Germany) was utilized to characterize the microstructures. The thermal diffusivity of the sintered samples was measured using a laser-flash technique (LFA-457, Laser Flash Thermal Constant Analyzer, Netzsch Co., Selb, Germany). The equation: k = ρ C_p_ α, where ρ, Cp, and α are the density, specific heat, and thermal diffusivity, respectively, was used to determine the thermal conductivity k. Additionally, Cp has a value of 0.68 J·g^−1^·K^−1^. The specification of the samples after being polished for testing was 10 × 10 × 2 mm^3^.

Sintered samples were cut and ground to test bars with dimensions of 2 × 3 × 14 mm^3^ to analyze the mechanical properties. Flexural strength was assessed using a three-point bending technique and a crosshead speed of 0.5 mm/min. The fracture toughness was evaluated using a single-edge notched beam method by applying test bars of 2 × 4 × 14 mm^3^. The crosshead speed was set to 0.05 mm/min, and the depth of the notch was 2 mm. A Vickers hardness tester (Tukon2500B, Wilson, Sioux City, IA, USA) was used to measure the Vickers hardness (HV) under 15 s of 5 kg loading.

## 3. Results and Discussion

### 3.1. SPS and UHP Sintering

Figure 3 shows the XRD diffraction of the sintered samples by SPS at 1550 °C for 3 min and UHP sintering at 1550 °C for 15 min. It demonstrates that the prominent diffraction peaks of sample SPS1 and SPS2 were β-Si_3_N_4_ with no trace of α-Si_3_N_4_. The fragile crystalline phase is Y_2_Si_3_O_3_N_4_. In contrast, the diffraction peaks of UHP1 and UHP2 are composed mainly of β-Si_3_N_4_, residual α-Si_3_N_4_, and the crystalline phase of YMgSi_2_O_5_N. The content of β-Si_3_N_4_ was calculated in Table 3. It can be seen that the content of β-Si_3_N_4_ in the SPS sintered samples all reached 92%, while the range of β-Si_3_N_4_ in the UHP sintered samples was lower, at about 87%.

It is known that in the process of ordinary pressure sintering, the densification and phase transition processes were almost carried out simultaneously. In the SPS sintering process, following the development of the liquid phase, the process of densification commenced, and the eutectic temperature of the MgO and SiO_2_ was approximately 1350 °C, which was slightly lower than that of the phase transition temperature of the silicon nitride. Therefore, the α-to-β phase transition was carried out entirely in 3 min during the solution-reprecipitation stage, especially with the low content of α-Si_3_N_4_ in the raw materials.

However, Shen et al. [26] revealed that the ultra-high pressure sintering process could be divided into two procedures. The first procedure was the bonding process between the powder particles to form a dense block with a specific hardness, and the second procedure was the sintering addition’s reaction with SiO_2_ on the surface of silicon nitride powder to form the liquid phase, which began the phase transition process. The sintering driving force of coarse β-Si_3_N_4_ powders was lower than fine powders, and it took a long time under the driving force of ultra-high pressure sintering. According to Shen et al. [27], silicon nitride initially tended to combine with rare earth oxides to lower the system energy rather than converting to β-Si_3_N_4_. As a result, UHP sintered samples showed significant diffraction peaks of the YMgSi_2_O_5_N crystalline secondary phase. Therefore, all of the above reasons have prolonged the α-to-β phase transition process, and the phase transition has not been carried out thoroughly.

The UHP sintered samples have a wider peak shape compared to the high and tight peak shape of the SPS sintered samples. The size of the grain and the crystallinity are indicated by the peak height and width of the XRD pattern. The grain size and crystallization quality increase with the narrower and higher peak; on the other hand, the grain size and crystallization quality may decrease with a broader and shorter peak [28].

The bulk densities and relative densities and β-phase content of samples sintered by SPS at 1550 °C for 3 min and UHP at 1550 °C for 15 min are exhibited in Table 3. It shows that the UHP sintered samples’ relative densities reached over 99.2%. The relative density of SPS sample SPS2 reached 99.2%, but the relative density of sample SPS1 slightly decreased to 98.5%. It is known that the forces responsible for driving densification during the SPS silicon nitride ceramics are external mechanical pressure and liquid-phase capillary force. Additionally, the liquid phase’s capillary force, which is produced when sintering aids and SiO_2_ react on the surface of Si_3_N_4_, is crucial for the particle rearrangement stage. Therefore, sample SPS1 has a slightly lower density than sample SPS2 because there was less MgO present, which prevented enough of the liquid phase from forming quickly enough to densify sample SPS1. Nevertheless, ultra-high pressure may be the dominant driving force for UHP sintering, and there may not be much of a difference in the density of UHP1 and UHP2 with various additions of MgO.

The SEM micrographs of the etched surface of the samples sintered by SPS and UHP at 1550 °C are shown in Figure 4. The microscopic morphology of the samples shows a small number of large grains (>2 μm) distributed among small submicron matrix grains. The difference is that the number of large-sized grains of the UHP samples is much smaller. Cracks can be seen in the large grains of the high-magnification images. Chang and Rhodes [29] investigated the high-pressure sintering of uranium carbide at 4.6 GPa, indicating that sliding of the grain boundaries and fragmentation are significant sintering mechanisms. The coarse β-Si_3_N_4_ powder employed in this study has a particle size of 1.8–2.7 μm, and big grains are cracked or crushed at a high pressure of 5 GPa. As a result, the UHP samples exhibit a greater number of small grains. Due to the inclusion of some fine α-Si_3_N_4_ powder, the UHP1 and SPS1 samples have more small grains than samples UHP2 and SPS2. Moreover, there is a small number of pores in Figure 4c compared to Figure 4d, which is consistent with the lower density of sample SPS1.

### 3.2. Heat Treatment

Figure 5 shows the XRD diffractions of sintered samples after heat treatment at 1750 °C for 3 h. It illustrates that the prominent diffraction peaks of sintered samples are solely β-Si_3_N_4_, indicating that α-to-β phase transformations were successful in all cases. During the heat treatment process, the grain boundary phase re-melted to form the liquid phase, and the residual fine α-Si_3_N_4_ grains dissolved into the re-melted liquid phase and re-precipitated on large β-Si_3_N_4_ grains. Aside from the primary diffraction peaks attributable to β-Si_3_N_4_, several minor peaks ascribed to the crystalline phase can also be seen. It was discovered that the crystalline phases were YMgSi_2_O_5_N.

Figure 6 displays the SEM micrographs of the etched surfaces of the four samples after heat treatment at 1750 °C for 3 h. It is observed from Figure 5a that a few abnormal grains are distributed in the fine matrix grains. According to Kong, J. H. [30], the driving force for aberrant grain growth may be dependent on the status of the effective space around a grain. During UHP sintering, many large grains were fragmented into small grains, leaving a small number of large grains as the nucleus, whereas the space around each nucleus was supersaturated. The residual α-Si_3_N_4_ and small β-Si_3_N_4_ grains dissolved in the liquid phase during heat treatment and re-precipitated on the surface of the large β-Si_3_N_4_ nucleus to promote the growth of abnormal grain. In Figure 5b, some large elongated grains were distributed among small rod-like matrix grains that formed an interlocking structure due to a greater number of large residual grains than the sample UHP1 after UHP sintering. On the contrary, it shows a microstructure with several grains greater than 2 μm in diameter distributed among the fine matrix grains in Figure 5c,d. Research [22,31] indicated that with the excessive addition of the nuclei, the effective growth spaces around the nuclei are insufficient and suppress the growth of grains due to the steric hindrance. After SPS, α-Si_3_N_4_ almost transformed completely into β-Si_3_N_4_ in the SPS samples, with excessive large grains as the nucleus leading to the suppression of grain growth, resulting in a microstructure with a smaller aspect ratio. Compared to sample SPS2, sample SPS1 contains more small-sized grains due to the addition of some fine α-Si_3_N_4_ powder to the raw material powder. The grain boundary phase is re-melted to form the filling of the liquid phase, and the Ostwald ripening of the β-Si_3_N_4_ grains promotes the elimination of the pores in sample SPS1 after heat treatment.

The bulk density, relative density, Vickers hardness, fracture strength, and fracture toughness after heat treatment was illustrated in Table 4. The mechanical properties of silicon nitride ceramics are mainly related to the α/β phase content, density, grain shape, second phase content, and distribution. High α-Si_3_N_4_ content, high density, and a high number of small-sized grains are beneficial to improve the hardness of silicon nitride ceramics. The bulk densities of all samples reached more than 99.4% after heat treatment, which was close to complete densification, with the highest densities of 99.8% for SPS1 samples. The α-Si_3_N_4_ was transformed entirely into β-Si_3_N_4_ after heat treatment; therefore, the number of small-sized grains is the main factor affecting the hardness of ceramics. The Vickers hardness of sample UHP1 reached a high of 1918 HV due to the mass amount of fine matrix grains. The Vickers hardness of samples UHP2 and SPS2 decreased with the increasing number of large grains. It is considered that larger-sized grains tend to introduce larger defects and cause a reduction in material strength. However, Hu [31] found that Si_3_N_4_ ceramics had both high bending strength and fracture toughness when the grain size distribution shows a “bimodal” distribution with a small number of large long rod-like grains uniformly distributed among the small grains. Sample UHP2 achieves the highest value of fracture strength (822 MPa) and fracture toughness (6.6 MPa·m^1/2^) because of the interlocking structure of toughened elongated grains. Due to the bimodal microstructure with some large grains distributed among small grains, the fracture strength (817 MPa) and fracture toughness (6.2 MPa·m^1/2^) of sample SPS1 is comparable to sample UHP2. Moreover, the fracture strength and fracture toughness of sample SPS2 decreased to 760 MPa and 6.0 MPa·m^1/2^ with the increased quantities of large homogeneous grains. Sample UHP1 has the lowest fracture strength of 569 MPa, and the fracture toughness of 5.14 MPa·m^1/2^ may account for the growth of a few abnormal grains, which were detrimental to the mechanical properties.

Table 5 shows the mean diameter, the area fraction of the grain boundary, thermal diffusivity, and thermal conductivity of the UHP and SPS sintered samples after heat treatment. The thermal conductivity of silicon nitride ceramics is influenced by density, grain size, content, and distribution of inter-granular secondary phase and lattice defects. Large grain can improve the heat transmission efficiency, and a low range of inter-granular secondary phase and lattice defects can reduce the phonon scattering to improve the thermal conductivity. It can be seen from Table 5 that the mean diameters of the SPS samples are more extensive than that of the UHP sintered samples. Therefore, the thermal diffusivity and thermal conductivity of SPS samples are higher than that of UHP samples. The UHP1 and SPS1 samples’ area fraction of the grain boundary is lower than that of samples UHP2 and SPS2 due to the decreased addition of MgO. Therefore, with the lower amount of grain boundary phase, the larger grain size, and the highest density, sample SPS1 reaches the highest thermal conductivity 71 W·m^−1^·K^−1^ compared with sample SPS2.

## 4. Conclusions

Dense Si_3_N_4_ ceramics were fabricated with coarse β-Si_3_N_4_ powders as raw materials by SPS at 1550 °C for 3 min and UHP at 1550 °C for 15 min, followed by heat treatment at 1750 °C for 3 h. Densification, phase transition, microstructure, mechanical properties, and thermal conductivities were investigated. The following conclusions were drawn from the present work.

(1)Compared with UHP sintering, SPS can fabricate dense Si_3_N_4_ ceramics with coarse β-Si_3_N_4_ powders as raw materials in a shorter time and promote the α-to-β phase transition;(2)Coarse β-Si_3_N_4_ grains were partially fragmented during ultra-high pressure sintering under high pressure in 5 GPa, thereby reducing the number of the nucleus, which is a benefit for the growth of the elongated grains. Therefore, sample UHP1 achieved the highest fracture strength (822 MPa) and fracture toughness (6.6 MPa·m^1/2^), due to the interlocking structure of toughened elongated grains grown from residual coarse β-Si_3_N_4_ grains during heat treatment. Due to the bimodal microstructure with some large grains distributed among small grains, the fracture strength (817 MPa) and fracture toughness (6.2 MPa·m^1/2^) of sample SPS1 was comparable to sample UHP2 due to the high density;(3)SPS samples achieved higher thermal conductivity on account of a larger mean diameter of grains compared with ultra-high pressure sintering samples. The high thermal conductivity of Sample SPS1 was 71 W·m^−1^·K^−1^ due to the high density and the reduced grain boundary phase.

## Figures and Tables

**Figure 1 materials-15-07309-f001:**
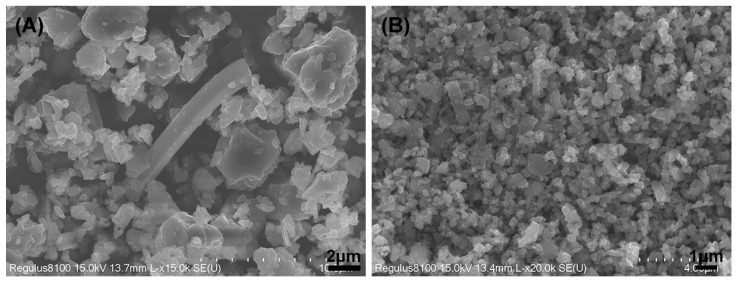
Morphology of raw material powders: (**A**) β-Si_3_N_4_ powder; (**B**) α-Si_3_N_4_ powder.

**Figure 2 materials-15-07309-f002:**
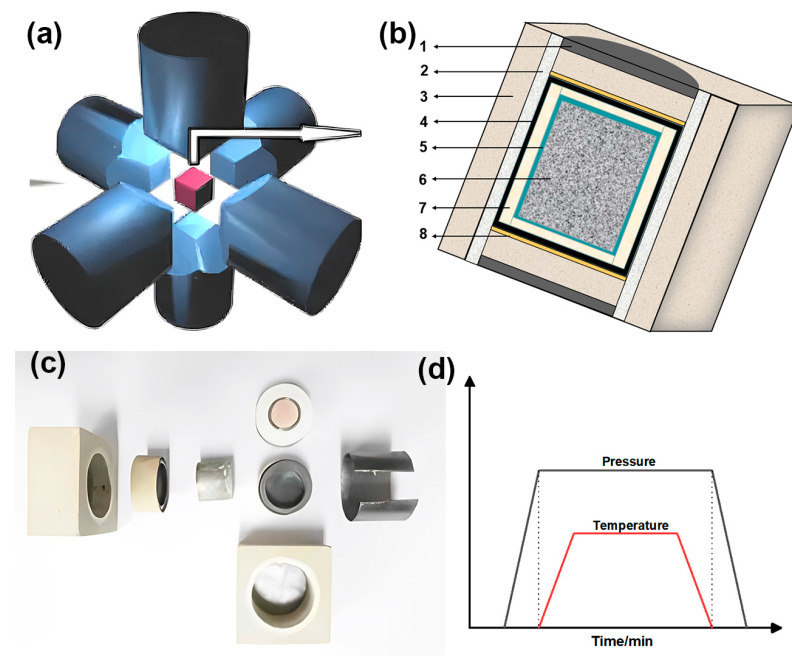
Schematic diagram of UHP sintering assembly: (**a**) Six-anvil pressure diagram; (**b**) Cross section diagram: 1: conductive steel hat, 2: dolomite, 3: pyrophyllite composite cavity, 4: graphite chips, 5: molybdenum cup, 6: raw mixed powders, 7: salt composite, 8: copper chip; (**c**) Physical disassembled image; (**d**) Sintering process curve.

**Figure 3 materials-15-07309-f003:**
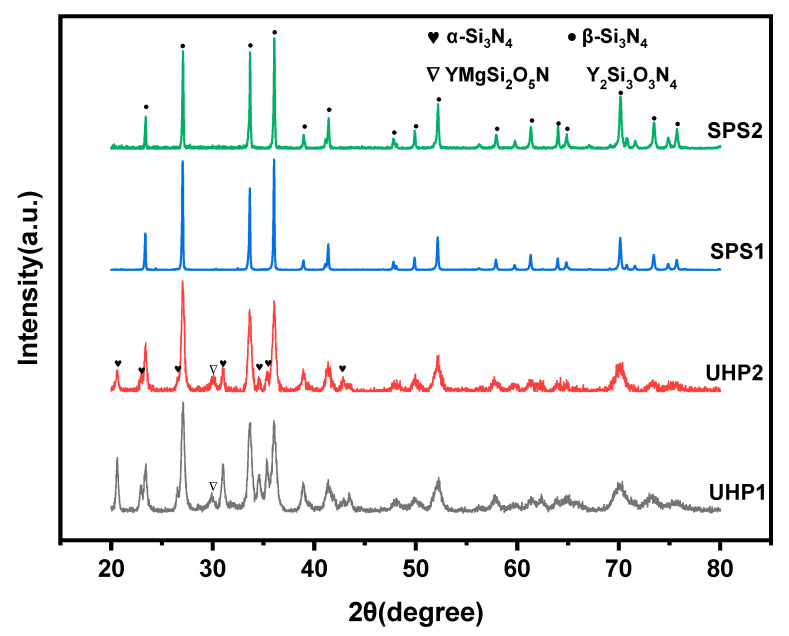
XRD diffractions of sintered samples by SPS at 1550 °C for 3 min and UHP at 1550 °C for 15 min.

**Figure 4 materials-15-07309-f004:**
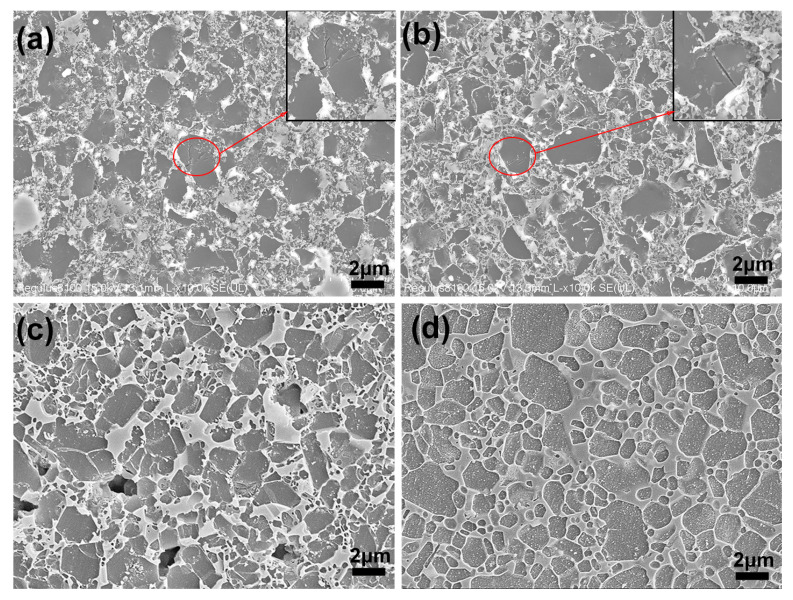
SEM images of the polished and plasma-etched surfaces of the Si_3_N_4_ ceramics sintered by SPS at 1550 °C for 3 min and UHP at 1550 °C for 15 min: (**a**) UHP1; (**b**) UHP2; (**c**) SPS1; (**d**) SPS2.

**Figure 5 materials-15-07309-f005:**
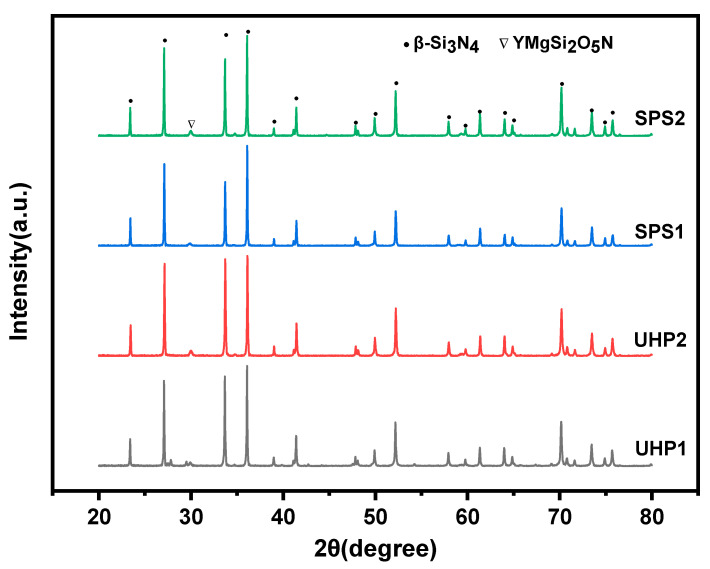
XRD diffractions of sintered samples after heat treatment at 1750 °C for 3 h.

**Figure 6 materials-15-07309-f006:**
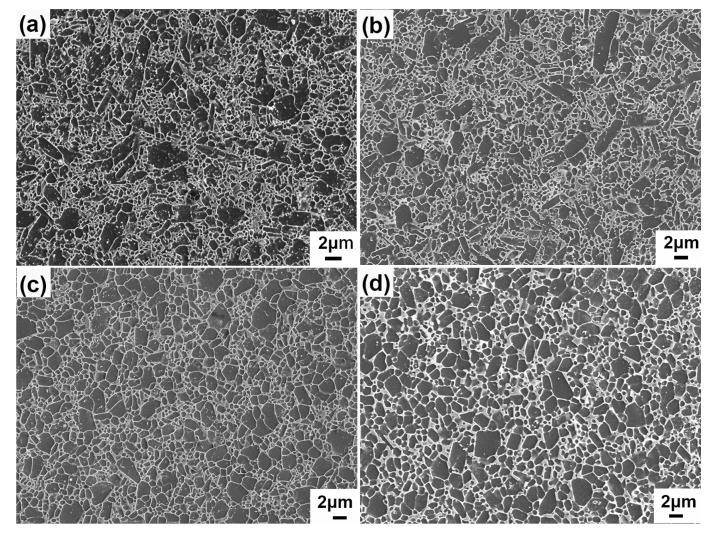
SEM images of the polished and plasma-etched surfaces of the Si_3_N_4_ ceramics after heat treatment at 1750 °C for 3 h: (**a**) UHP1; (**b**) UHP2; (**c**) SPS1; (**d**) SPS2.

**Table 1 materials-15-07309-t001:** The properties of the raw Si_3_N_4_ powders.

Property	β-Si_3_N_4_	α-Si_3_N_4_
α-phase content/mass%	18	>90
Oxygen/mass%	<0.8	<2.2
Aluminum/ppm	<3	<400
Calcium/ppm	<3	<400
Iron/ppm	<3	<50
D_50_/μm	1.8–2.7	0.8

**Table 2 materials-15-07309-t002:** Composition of the starting powder (wt.%) and sintering parameter.

Specimens	Composition in Mass/%				Sintering Condition		
	β-Si_3_N_4_	α-Si_3_N_4_	MgO	Y_2_O_3_	Sintering Temperature (°C)	Holding Time (min)	Pressure
UHP1	76	18	3	3	1550	15	5 GPa
UHP2	92		5	3	1550	15	5 GPa
SPS1	76	18	3	3	1550	3	40 MPa
SPS2	92		5	3	1550	3	40 MPa

**Table 3 materials-15-07309-t003:** Bulk densities, relative densities, and β-phase content of samples after SPS and UHP sintering.

Specimens	Bulk Density (g cm^−3^)	Relative Density (%)	β-Phase Content
UHP1	3.211	99.2	86.96%
UHP2	3.218	99.3	87.85%
SPS1	3.187	98.5	99.47%
SPS2	3.214	99.2	92%

**Table 4 materials-15-07309-t004:** Bulk density, relative density, Vickers hardness, fracture strength, and fracture toughness of the samples after heat treatment.

Specimens	Bulk Density (g cm^−3^)	Relative Density (%)	Vickers Hardness (HV)	Fracture Strength (MPa)	Fracture Toughness (Mpa·m^1/2^)
UHP1	3.216	99.4	1918	569	5.6
UHP2	3.221	99.4	1709	822	6.6
SPS1	3.230	99.8	1890	817	6.2
SPS2	3.223	99.5	1367	760	6.0

**Table 5 materials-15-07309-t005:** Mean diameter, the area fraction of grain boundary, and thermal properties of the samples after heat treatment.

Specimens	Mean Diameter (μm)	Area Fraction of Grain Boundary (Area %)	Thermal Diffusivity (mm s^−1^)	Thermal Conductivity (W·m^−1^·K^−1^)
UHP1	0.6	36	20.175	44
UHP2	0.8	39	22.314	49
SPS1	0.9	33	32.384	71
SPS2	1.0	40	26.455	58

## Data Availability

Data sharing is not applicable to this article.

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
