# Peer review of "Effect of Ultra-High Pressure Sintering and Spark Plasma Sintering and Subsequent Heat Treatment on the Properties of Si3N4 Ceramics"

_materials, 2022, doi:10.3390/ma15207309_

Round 1
Reviewer 1 Report
The work presented has interesting aspects in the materials area. However, some modifications are necessary to improve the text as shown below:
Update references;
Would it be possible to add a table comparing the results obtained with other studies in the literature?;
Check the English of the text;
The results and discussion section should be imprinted with more relevant discussions;
Could the authors cite possible applications that the technique used would allow?;
This information could be added both in the results and discussion section as well as in the conclusion;
Author Response
Dear Reviewer,
Thank you very much for your time involved in reviewing the manuscript and your very encouraging comments on the merits.
Comments:
The work presented has interesting aspects in the materials area. However, some modifications are necessary to improve the text as shown below:
- Update references;
Reply: Thank you very much for your notices. We have updated the references and most of the references are from the last 5 years, but there are still a few articles on the preparation of silicon nitride ceramics by ultra-high pressure that have been published more than 10 years ago, since there are fewer articles using the method of ultra-high pressure to prepare silicon nitride ceramics in the last 5 years.
- Would it be possible to add a table comparing the results obtained with other studies in the literature?;
Reply: Thank you for your advices. Studies on the preparation of silicon nitride ceramics by ultra-high pressure sintering have mainly focused on phase transition and hardness, and there are few studies on other mechanical and thermal properties, so there are few relevant research data. Therefore, the results of this work were not compared with other literature results in the manuscript. Among the articles using ultra-high pressure to prepare silicon nitride ceramics in the last five years, only Li et al. prepared silicon nitride ceramics by ultra-high pressure sintering and tested the thermal conductivity in 2018, and the literature has been cited in the introduction.
- Check the English of the text;
Reply: Thank you for your valuable comments. We have carefully and thoroughly proofread the manuscript to correct all the grammar and typos.
- The results and discussion section should be imprinted with more relevant discussions;
Reply: Thank you for your helpful suggestions. We have supplemented the discussion section by adding theoretical descriptions and relevant studies, which we hope will help readers better understand the experimental results. All the changes have been highlighted in red in the revised manuscript.
- Could the authors cite possible applications that the technique used would allow?;
This information could be added both in the results and discussion section as well as in the conclusion;
Reply: Thank you for your advice. In practical applications, UHP sintering is mainly used to prepare super-hard materials such as diamond and PCD tools. UHP sintering in ceramics is mainly used for the preparation of ultra-high temperature ceramics, and the related literatures are cited in the introduction. SPS sintering is more widely used in laboratory research for the preparation of metals, ceramics, and composites, and the related literatures are cited in the introduction.
We would like to take this opportunity to thank you for all your time involved and this great opportunity for us to improve the manuscript. We hope you will find this revised version satisfactory.
Sincerely,
Xiaoan Lv

Reviewer 2 Report
Reviewer Comment
The paper stated on “Comparative study of the effects of ultra-high pressure sintering and spark plasma sintering and heat-treatment on the properties of silicon nitride ceramics”
1. The title of paper is too long and difficult to understand. It should be revised.
2. The ‘’Abstract section’’ needs to improve more. It is difficult to find the main highlight of this research. Also, the error can be found in this section, such as β-Si3N4 (on line: 20).
3. On line: 22, The 21 sample S1 obtained the best….What is the sample S1? It’s not clear.
4. Why the author should Bold(Ctrl+B) for the keywords? (On line: 25-26).
5. In the section “Materials and Methods” Where is the image of High purity β-Si3N4 and α-Si3N4 powder?
6. It is very difficult to understand about Figure. 1. The images and 3D schematic design that related to the “Methods” used in this paper should be added.
7. The quality of this paper needs to be improved. Especially, on section 2. “Materials and Methods”. It is unacceptable due to lack of explanation.

Author Response
Dear Reviewer,
Thank you very much for your time involved in reviewing the manuscript and your very encouraging comments on the merits.
Comments:
The paper stated on “Comparative study of the effects of ultra-high pressure sintering and spark plasma sintering and heat-treatment on the properties of silicon nitride ceramics”
- The title of paper is too long and difficult to understand. It should be revised.
Reply: Thank you for your helpful suggestions. We revised the title of paper to “Effect of ultra-high pressure sintering and spark plasma sintering and subsequent heat treatment on the properties of Si3N4 ceramics”. And we are expecting the revised title to be more conducive to understanding.
- The ‘’Abstract section’’ needs to improve more. It is difficult to find the main highlight of this research. Also, the error can be found in this section, such as β-Si3N4(on line: 20).
Reply: Thank you for your great suggestion on improving the “Abstract section”. We have modified the abstract section to emphasize that the fragment of UHP is beneficial to the mechanical properties and the addition of fine powder is beneficial to the SPS samples to obtain the optimal comprehensive properties. The modified parts are also marked in red, and the relevant contents are provided below as a screen dump for your quick reference.
- On line: 22, The 21 sample S1 obtained the best….What is the sample S1? It’s not clear.
Reply: Thank you for your valuable comments. The sample S1is the β-Si3N4 ceramics SPS sintered with fine α-Si3N4 powder addition as shown in Table 2. And we replace “The sample S1” with “The β-Si3N4 ceramics SPS sintered with fine α-Si3N4 powder addition”. The relevant contents are provided below as a screen dump for your quick reference.
Table 2. Composition of starting powder (wt.%) and sintering parameter.
|
Specimens |
Composition in mass/% |
Sintering condition |
|||||
|
β-Si3N4 |
α-Si3N4 |
MgO |
Y2O3 |
Sintering temperature(°C) |
Holding time (min) |
Pressure |
|
|
U1 |
76 |
18 |
3 |
3 |
1550 |
15 |
5GPa |
|
U2 |
92 |
|
5 |
3 |
1550 |
15 |
5GPa |
|
S1 |
76 |
18 |
3 |
3 |
1550 |
3 |
40MPa |
|
S2 |
92 |
|
5 |
3 |
1550 |
3 |
40MPa |
- Why the author should Bold (Ctrl+B) for the keywords? (On line: 25-26).
Reply: Thank you for the detailed review. It was an oversight on my part not to double check the bolded keywords. We have revised the keyword section, and the relevant contents are provided below as a screen dump for your quick reference.
- In the section “Materials and Methods” Where is the image of High purity β-Si3N4and α-Si3N4powder?
Reply: Thank you for the meticulous review. We have added the image of high purity β-Si3N4 and α-Si3N4 powder, as shown in Figure 1.
Figure 1. Morphology of raw material powders: (A) β-Si3N4 powder; (B) α-Si3N4 powder
- It is very difficult to understand about Figure. 1. The images and 3D schematic design that related to the “Methods” used in this paper should be added.
Reply: Thank you for your valuable comments. We have drawn a new 3D schematic diagram of the cross section and added a schematic diagram of the six-sided pressure device, the physical disassembled image and the sintering process curve as shown in Figure 2. And the UHP sintering methods is explained in detail for easy understanding. The relevant contents are provided below as a screen dump for your quick reference.
Figure 2. Schematic diagram of UHP sintering assembly: (a) Six-anvil pressure diagram; (b)Cross section diagram: 1: conductive steel hat, 2: dolomite,3: pyrophyllite composite cavity, 4: graphite chips, 5: molybdenum cup, 6: raw mixed powders, 7: salt composite, 8: copper chip; (c) Physical disassembled image; (d)Sintering process curve
- The quality of this paper needs to be improved. Especially, on section 2. “Materials and Methods”. It is unacceptable due to lack of explanation.
Reply: Thanks for your helpful advice. We have carefully and thoroughly proofread the manuscript to improve the quality. Especially the section 2. “Materials and Methods”, we added morphology of raw material powders and schematic diagram of UHP sintering assembly, and described the experimental method in detail. The revised parts are also highlighted in red for easy viewing.
We would like to take this opportunity to thank you for all your time involved and this great opportunity for us to improve the manuscript. We hope you will find this revised version satisfactory.
Sincerely,
Xiaoan Lv

Reviewer 3 Report
Dear Authors,
You have done a great job!
The flow of writing is good. The comparison between the effects of two different sintering techniques on the mechanical and thermal properties of Si3N4 ceramic powders was made successfully.
There are several aspects needs improvement:
1-From the abstract:
Line 12: Coarse Beta silicon nitride (β-Si3N4) powder were used as raw materials to fabricate dense Si3N4
20Throughout the manuscript, the abbreviations should be specified from the beginning (for example at the introduction section), for example Spark Plasma Sintering (SPS), ....etc. These abbreviations should be used after that without need to repeat the full name of methodologies, ..etc.
3- Line 60: It was reported that Si3N4 ceramics w...
4-Line 157: of α-Si3N4 in raw materials was low
Note: these subscriptions should be used correctly.
5- Passive voice should be used instead of active voice. just for example,
Line 183: As we know, ???
Can be changed to: from the literature, ....
or: As it is known,...
Note: All of these active voice should be modified to passive voice throughout the manuscript to adequate scientific soundness in academia.
6- SEM micrographs, Figure 3 and Figure 5
they should be labelled as Figure 3a, Figure 3b, ... instead of Figure 3 S1, Figure 3 U2, and same for Figure 5.
7- English language needs improvement.
Just few examples:
Line 186: sintering Si3N4, so the density.....
this can be rewritten as the following: sintering Si3N4, hence, the density
Line 203: Therefore more small grains can be seen...
Therefore, more....
Punctuation is needed to be considered.
8-The discussion of Table 4, from line 243 to line 257, needs improvement and to be clearer.
9- line 247: Hv ..HV
10- in the methodology, the authors did not mention the method of hardness measurement, and instrument used for hardness measurement?
11- in the abstract, line 22: instead of writing S1, the authors can write:
the β-Si3N4 SPS sintered added with α-Si3N4 obtained the best comprehensive performance,...
Author Response
Dear Reviewer,
Thank you very much for your time involved in reviewing the manuscript and your very encouraging comments on the merits.
Comments:
You have done a great job!
The flow of writing is good. The comparison between the effects of two different sintering techniques on the mechanical and thermal properties of Si3N4 ceramic powders was made successfully.
There are several aspects needs improvement:
1-From the abstract:
Line 12: Coarse Beta silicon nitride (β-Si3N4) powder were used as raw materials to fabricate dense Si3N4
Reply: Thank you for your professional advice. This expression is more in conformity with the specifications of English writing, and we have modified this part as shown below.
2-Throughout the manuscript, the abbreviations should be specified from the beginning (for example at the introduction section), for example Spark Plasma Sintering (SPS), ....etc. These abbreviations should be used after that without need to repeat the full name of methodologies, ..etc.
Reply: Thank you for your helpful advice. We have specified the abbreviations from the introduction section, and used the abbreviations instead of repeating the full name in the following sections. The relevant contents are provided below as a screen dump for your quick reference.
3- Line 60: It was reported that Si3N4 ceramics w...
Reply: Thank you for your specialized advice. We revised the sentence as follows:
4-Line 157: of α-Si3N4 in raw materials was low
Note: these subscriptions should be used correctly.
Reply: Thank you for your professional suggestions. Such grammatical errors are things I should avoid, and I will check more carefully for grammatical correctness in my future writing. The sentence is revised as follows:
5- Passive voice should be used instead of active voice. just for example,
Line 183: As we know, ???
Can be changed to: from the literature, .... or: As it is known,...
Note: All of these active voice should be modified to passive voice throughout the manuscript to adequate scientific soundness in academia.
Reply: Thank you for your professional advice. I will improve my English writing and focus on passive sentence practice in my future writing. We checked the active voice throughout the manuscript and modified them to passive voice. All the changes have been highlighted in red in the revised manuscript.
6- SEM micrographs, Figure 3 and Figure 5
they should be labelled as Figure 3a, Figure 3b, ... instead of Figure 3 S1, Figure 3 U2, and same for Figure 5.
Reply: Thank you for the attentive review. We have modified the labels in Figure 5 as you suggested, as shown below.
Figure 4. SEM images of the polished and plasma-etched surfaces of the Si3N4 ceramics sintered by SPS at 1550°C for 3 min and UHP at 1550°C for 15 min: (a)U1; (b)U2; (c)S1; (d)S2
Figure 6. SEM images of the polished and plasma-etched surfaces of the Si3N4 ceramics after heat treatment at 1750 °C for 3h: (a)U1; (b)U2; (c)S1; (d)S2
7- English language needs improvement.
Just few examples:
Line 186: sintering Si3N4, so the density.....
this can be rewritten as the following: sintering Si3N4, hence, the density
Line 203: Therefore more small grains can be seen...
Therefore, more....
Punctuation is needed to be considered.
Reply: We really appreciate your suggestions. English writing is my weakness and I have been trying to improve my writing skills for a long time. I will pay more attention to punctuation and conjunctions in my future writing. We have carefully checked the article several times to minimize errors in grammar and punctuation. All the changes have been highlighted in red in the revised manuscript.
8-The discussion of Table 4, from line 243 to line 257, needs improvement and to be clearer.
Reply: Thanks for your professional advice. We have improved the discussion of Table 4 by adding relevant theoretical studies and corresponding research results, and by analyzing the results in more details in order to express them more clearly. The relevant contents are provided below as a screen dump for your quick reference.
9- line 247: Hv ..HV
Reply: Thank you for your careful review. I apologize for this minor error in the manuscript and have corrected it.
10- in the methodology, the authors did not mention the method of hardness measurement, and instrument used for hardness measurement?
Reply: We are very grateful for your careful review of the manuscript. It is my negligence not to mention the method of hardness measurement and instrument used for hardness measurement. In the revised manuscript we have added the method of hardness measurement and instrument used for hardness measurement. The relevant contents are provided below as a screen dump for your quick reference.
11- in the abstract, line 22: instead of writing S1, the authors can write:
the β-Si3N4 SPS sintered added with α-Si3N4 obtained the best comprehensive performance,...
Reply: Thank you very much for your detailed review. The use of abbreviations in abstracts does make it difficult for others to understand their exact meaning, and your comments will enable readers to better understand the content of the article. We have made revisions to the relevant sentences in the abstract. The relevant contents are provided below as a screen dump for your quick reference.
We would like to take this opportunity to thank you for all your time involved and this great opportunity for us to improve the manuscript. We hope you will find this revised version satisfactory.
Sincerely,
Xiaoan Lv

Round 2
Reviewer 1 Report
Dear editor, please consider the article as accepted.
Author Response
Dear Reviewer,
Thank you very much for your time in reviewing the manuscript and for recognizing my article.
Sincerely,
Xiaoan Lv

Reviewer 2 Report
The quality of paper is improved.
It can be accepted.
Author Response

(The authors gave the same response as above.)
